# Subcritical Water Extraction of *Salvia miltiorrhiza*

**DOI:** 10.3390/molecules26061634

**Published:** 2021-03-15

**Authors:** Brahmam Kapalavavi, Ninad Doctor, Baohong Zhang, Yu Yang

**Affiliations:** 1Department of Chemistry, East Carolina University, Greenville, NC 27858, USA; brahmam.kapalavavi@pfizer.com (B.K.); ninad11@hotmail.com (N.D.); 2Department of Biology, East Carolina University, Greenville, NC 27858, USA; zhangb@ecu.edu

**Keywords:** active pharmaceutical ingredients, reproduction, medicinal herbs, *Salvia miltiorrhiza*, subcritical water extraction

## Abstract

In this work, a green extraction technique, subcritical water extraction (SBWE), was employed to extract active pharmaceutical ingredients (APIs) from an important Chinese medicinal herb, *Salvia miltiorrhiza* (danshen), at various temperatures. The APIs included tanshinone I, tanshinone IIA, protocatechualdehyde, caffeic acid, and ferulic acid. Traditional herbal decoction (THD) of *Salvia miltiorrhiza* was also carried out for comparison purposes. Reproduction assay of herbal extracts obtained by both SBWE and THD were then conducted on *Caenorhabditis elegans* so that SBWE conditions could be optimized for the purpose of developing efficacious herbal medicine from *Salvia miltiorrhiza.* The extraction efficiency was mostly enhanced with increasing extraction temperature. The quantity of tanshinone I in the herbal extract obtained by SBWE at 150 °C was 370-fold higher than that achieved by THD extraction. Reproduction evaluation revealed that the worm reproduction rate decreased and the reproduction inhibition rate increased with elevated SBWE temperatures. Most importantly, the reproduction inhibition rate of the SBWE herbal extracts obtained at all four temperatures investigated was higher than that of traditional herbal decoction extracts. The results of this work show that there are several benefits of subcritical water extraction of medicinal herbs over other existing herbal medicine preparation techniques. Compared to THD, the thousand-year-old and yet still popular herbal preparation method used in herbal medicine, subcritical water extraction is conducted in a closed system where no loss of volatile active pharmaceutical ingredients occurs, although analyte degradation may happen at higher temperatures. Temperature optimization in SBWE makes it possible to be more efficient in extracting APIs from medicinal herbs than the THD method. Compared to other industrial processes of producing herbal medicine, subcritical water extraction eliminates toxic organic solvents. Thus, subcritical water extraction is not only environmentally friendly but also produces safer herbal medicine for patients.

## 1. Introduction

Due to its green nature and low side effects, herbal medicine has gained greater attention in the Western world nowadays [1,2,3]. Both raw and preprepared herbal medicines are available in many developed countries [4,5].

The traditional way for patients to take the herbal medicine prescribed by doctors is to cook the medicinal herbs in boiling water for 60 to 90 min and then drink the “soup medicine”. This herbal medicine preparation method is called traditional herbal decoction (THD). Although this herbal decoction method has been used since ancient times, there are several major drawbacks associated with it. Firstly, a large portion of the volatile active pharmaceutical ingredients (APIs) contained in medicinal herbs are lost during the cooking process with boiling water. This is because the decoction process is an open system, and volatile APIs are thus lost to the atmosphere as vapor. Secondly, some APIs contained in medicinal herbs may be degraded due to the prolonged cooking time of 60 to 90 min. In both cases, the effectiveness of THD extracts, the soup medicine, in treating diseases may be decreased due to the reduced quantity of APIs in the herbal medicine as a result of them being lost to vapor or by degradation. Concurrently, even as APIs are lost, compounds with detrimental health effects may be extracted during the lengthy THD process. The presence of such toxicants in medicinal herbal extracts may not be safe for patient use. Lastly, it would be a rare coincidence for 100 °C to be the best temperature for effective extraction of all APIs from medicinal herbs. Proper scientific investigation of other temperatures may yield more potent yet safer herbal medicine.

Several other methods have been used for extraction of herbs and plants, including Soxhlet extraction, sonication, pressurized liquid extraction, accelerated solvent extraction, microwave-assisted extraction, and sub- and supercritical fluid extraction [6,7,8,9,10]. Because organic solvents are used in most of these extraction techniques, such as Soxhlet and sonication extractions, they are not suitable for preparing herbal medicine due to the toxicity of organic solvents.

Herbal extracts, such as small bags of medicinal herb extracts, are prepared by large-scale THD for patients so that they can take them directly without having to cook the herbs. This preprepared herbal medicine has gained popularity due to its convenience. Other forms of preprepared herbal medicines, such as tablets, capsules, and instant beverages, are also available commercially. While these products provide convenience to the consumer, their production via commonly used industrial extraction techniques is taking its toll on the environment and perhaps even on the patients. These techniques include maceration, vertical or turbo extraction, ultrasonic extraction, percolation, and counter current extraction. Many of the organic solvents required for use in these herbal extraction methods are toxic, and some are even carcinogenic [11]. The solvents required in these herbal preparation processes are costly not only to purchase but also for its waste disposal. Overall, such harsh extraction methods carry risks for the consumer and the environment, making them principally at odds with the perceived desire of the consumer who is likely looking for natural remedies rather than pollution-causing industrial processes and persistent trace carcinogens.

A scientifically rigorous path for modernization of herbal medicine preparation techniques is of great interest. It is important to not simply mimic THD but also to improve the efficacy of herbal medicines than those prepared by THD. This leads to this research, subcritical water extraction (SBWE) of medicinal herbs. Subcritical water refers to high-temperature and high-pressure water under conditions lower than the critical point of water: 374 °C and 218 atm. Water at elevated temperatures acts like an organic solvent due to its weakened hydrogen bonds and decreased polarity [12,13]. The solubility of organic compounds such as APIs in medicinal herbs is dramatically enhanced by simply increasing the water temperature. This unique characteristic of high-temperature water makes it an alternative mobile phase solvent for reversed-phase liquid chromatography [13,14,15,16] and an excellent extraction fluid for efficient removal of organics from various sample matrices, including plants and medicinal herbs [17,18,19,20,21,22,23]. Because different temperatures can be employed to carry out subcritical water extractions, there will be an optimized temperature that yields the highest quantity of APIs and in turn produces the most potent herbal medicine. Ideally, the solvent for extraction of medicinal herbs should be nontoxic, and the extraction technique should be more efficient in extracting active pharmaceutical ingredients and not cause their significant loss during the extraction process. Thus, subcritical water is an excellent choice for preparing herbal medicines.

In order to evaluate and optimize the SBWE technique, *Salvia miltiorrhiza* (also known as danshen in Chinese), a popular and important herb prescribed in traditional Chinese medicine (TCM), was used in this study. *Salvia miltiorrhiza* is a perennial plant in the genus *Salvia* of the mint family. Its roots are highly valued in traditional Chinese medicine and used in the treatment of various diseases, such as blood circulation, cardiovascular, and hepatic diseases [24,25,26]. Researchers have isolated about 70 compounds from the extract of *Salvia miltiorrhiza* [27]. Some of the identified anticancer compounds present in *Salvia miltiorrhiza* include tanshinone I, tanshinone IIA, protocatechualdehyde, caffeic acid, and ferulic acid. These APIs have already been found to demonstrate antiproliferative effect on various cancer cells, such as colon, leukemia, lung, and breast cancers, at either pre-clinical or clinical level [28,29,30,31]. Therefore, these five APIs were investigated in this study.

The main goal of this work was to investigate a potential herbal medicine preparation technique using subcritical water to yield efficacious herbal medicine. Therefore, subcritical water extraction of *Salvia miltiorrhiza* roots was carried out at four different temperatures (75, 100, 125, and 150 °C). For comparison and evaluation purposes, traditional herbal decoction of *Salvia miltiorrhiza* was also conducted. Then, these herbal extracts were characterized using GC/MS and HPLC to identify and quantify various anticancer agents. In order to evaluate the efficacy of the SBWE herbal extracts at various temperature conditions, the reproduction assay of SBWE and THD herbal extracts were conducted on *Caenorhabditis elegans*.

Despite being a simple multicellular organism, *Caenorhabditis elegans* has been widely employed to study complex behavior and syndromes. It has been used in many recent studies to understand human diseases, including cancer, ageing, development, addiction, and neurodegenerative diseases, as well as in pharmaceutical and toxicity studies [32,33,34,35,36]. Research on the worm bridges the gap between in vitro systems and preclinical studies in mammalian models. Experiments using cell lines often do not represent organism-level responses. On the other hand, *Caenorhabditis elegans* is particularly useful for reverse genetic approaches due to its short life cycle, availability of strains and feasibility of customized mutants, ability to perform complex behavior, and transparent cuticle for imaging assays. In this work, we employed *Caenorhabditis elegans* as a model system to investigate the drug potency of the extracted APIs from *Salvia miltiorrhiza* using reproduction analysis.

It is a novel approach to employ subcritical water for extraction of active pharmaceutical ingredients from medicinal herbs. Compared to THD, the thousand-year-old and yet still popular herbal preparation method, subcritical water extraction is conducted in a closed system. Therefore, no loss of volatile active pharmaceutical ingredients occurs due to loss of APIs to the open environment. However, loss of analytes can still occur due to degradation at elevated temperatures. Under optimized temperature, SBWE is more efficient in extracting APIs from medicinal herbs than the THD method. Compared to other industrial processes of producing herbal medicine, subcritical water extraction eliminates toxic organic solvents. Therefore, subcritical water extraction is not only more efficient and cheaper but also environment friendly because of its green nature. Many researchers have made efforts in recent years to develop greener analytical chemistry techniques. For example, an analytical Eco-Scale has been proposed as a tool for green analysis evaluation [37]. Another new tool introduced for assessment of the green character of analytical procedures is the Green Analytical Procedure Index [38]. The work reported in this paper also contributes to the field of green analytical chemistry.

## 2. Results and Discussion

### 2.1. Subcritical Water Extraction of Salvia Miltiorrhiza

As stated later in the Materials and Methods section, the quantification of all five APIs was achieved using a standard HPLC method. The concentration of the calibration solutions ranged from 0.002 to 1.00 mg/mL. The detection limit was 0.0002 mg/mL. The correlation coefficient (*r^2^*) ranged from 0.999 to 1.00.

A recovery study on the SBWE method was conducted using spiked samples (known amount of APIs) to validate the homemade SBWE system. The recoveries of the five APIs investigated in this work ranged from 95 to 102%, similar to that achieved in our previous study on SBWE of vanillin and coumarin [17]. This shows that the SBWE system is reliable. The subcritical extraction of *Salvia miltiorrhiza* was conducted at four different temperatures of 75, 100, 125, and 150 °C. Then, the SBWE extracts were characterized using GC/MS. Various analytes in the herbal extracts were identified by GC/MS by matching both GC retention times and mass spectra of standard samples. Among the identified analytes, five of them were anticancer agents: protocatechualdehyde, caffeic acid, ferulic acid, tanshinone I, and tanshinone IIA. Figure 1 shows the elution of the five compounds with an internal standard on GC/MS.

Figure 2 shows the HPLC separation of a standard solution (Figure 2a), methylene chloride phase after liquid–liquid extraction of an herbal extract obtained by SBWE at 125 °C (Figure 2b), and water phase (methanol was added) of an herbal extract obtained by SBWE at 125 °C (Figure 2c). As one can see, all five analytes and the internal standard were well separated.

Table 1 shows the quantification results of the five analytes present in the SBWE herbal extracts obtained at four different temperatures of 75, 100, 125, and 150 °C. The quantification results indicate that the protocatechualdehyde quantity extracted increased by 2-fold with the increase of extraction temperature from 75 to 100 °C and by 24-fold with further increase of extraction temperature from 100 to 125 °C. Then, with further increase of temperature from 125 to 150 °C, the extracted protocatechualdehyde quantity was enhanced 2.5-fold. There was no significant temperature effect on extraction efficiency of caffeic acid in the temperature range of 75 to 125 °C. However, the caffeic acid quantity found in the extract decreased at 150 °C due to possible degradation at such a high temperature. Ferulic acid was not detected at 75 and 100 °C, while its quantity extracted was improved by 32-fold when the temperature increased from 125 to 150 °C. The extraction efficiency, measured by analyte concentration in herbal extracts, of the two tanshinone compounds was clearly enhanced with increasing temperature, as shown in Table 1.

Table 1 also includes the quantities of the five analytes found in the THD extracts. One can easily see that tanshinone concentrations obtained by SBWE at all temperatures were much higher than those achieved by THD extractions. Specifically, tanshinone I concentration achieved by SBWE at 150 °C was 370-fold higher than that obtained by THD extraction, as demonstrated in Table 1.

We conducted the *t*-test on API concentrations in both THD and SBWE extracts. Our statistical analysis revealed that tanshinone I and tanshinone IIA concentrations obtained by THD at 100 °C and by SBWE at all four elevated temperatures were significantly different beyond the 99.9% confidence level. While there were no differences between caffeic acid concentrations achieved by THD at 100 °C and SBWE at 75–125 °C, the concentrations of caffeic acid obtained by THD and SBWE at 150 °C were significantly different at the 99.5% confidence level. Protocatechualdehyde concentrations achieved by THD at 100 °C and SBWE at all four temperatures were significantly different, mostly beyond the 99% confidence level.

### 2.2. Reproduction Assay of Caenorhabditis Elegans

First, we studied the impact of different concentrations (2, 10, and 50 times dilution with deionized water) of the herbal extract on *Caenorhabditis elegans* mortality. *Salvia miltiorrhiza* herbal extract was obtained by SBWE at 150 °C. After 30 h exposure to the three different diluted SBWE herbal extracts, the 10 times diluted herbal extract showed higher mortality rate than the other diluted herbal extracts. Therefore, the 10 times dilution factor was chosen for the remainder of the reproduction study. The API concentrations used for the reproduction assay are given in Table 2.

In order to optimize the preparation conditions of efficacious herbal medication through subcritical water extraction, the reproduction inhibition of the SBWE herbal extracts obtained at four different temperatures (75, 100, 125, and 150 °C) was evaluated on *Caenorhabditis elegans*. All SBWE herbal extracts were diluted 10 times with deionized water. Table 3 shows the reproduction assay of *Caenorhabditis elegans* after 30 h exposure to the 10 times diluted SBWE water extracts obtained at 75 to 150 °C. The reproduction inhibition of the extracts increased with higher extraction temperature. The SBWE extraction temperature also influenced mortality, as shown in Table 3. In general, the worm survival rate decreased with the increase in extraction temperature except at 150 °C. The main reason for lower mortality of *Caenorhabditis elegans* with the herbal extract obtained at 150 °C may be attributed to the less intake of highly concentrated herbal extract through the skin of *Caenorhabditis elegans*. Another reason for the lower mortality of worms may be due to the degradation of compounds associated with the mortality of worms, such as caffeic acid, at 150 °C.

The reproduction assay of *Caenorhabditis elegans* was also carried out using the THD extract of *Salvia miltiorrhiza* for comparison purposes. Both reproduction inhibition and mortality achieved by SBWE extracts at 100 °C and above were higher than those obtained by traditional herbal decoction, as shown in Table 3. The reproduction inhibition results indicate that SBWE is a much more efficient extraction technique than traditional herbal decoction, and it may be used to develop efficacious herbal medicine in the future.

We also carried out the *t*-test on reproduction assay results. Our statistical analysis showed that there was no difference in average reproduction and reproduction inhibition between the herbal extracts obtained by THD at 100 °C and SBWE at 75 °C, while there were significant differences between the extracts of THD 100 °C and SBWE at the other three elevated temperatures, mostly beyond the 99.9% confidence level. There were significant differences in mortality for the herbal extracts obtained by THD at 100 °C and SBWE at all four temperatures, mostly beyond the 99% confidence level.

## 3. Materials and Methods

### 3.1. Reagents and Supplies

Tanshinone I and tanshinone IIA were obtained from LKT Laboratories, Inc. (St. Paul, MN, USA). Protocatechualdehyde, caffeic acid, ferulic acid, sodium chloride, sodium hydroxide, agar, cholesterol, calcium chloride, calcium chloride dehydrate, and sodium phosphate dibasic heptahydrate were purchased from Sigma Aldrich (St. Louis, MO, USA). Sand, peptone, tryptone, magnesium sulfate, and magnesium sulfate heptahydrate were acquired from Fisher Scientific (Fair Lawn, NJ, USA). Potassium phosphate, dipotassium phosphate, yeast extract, and HPLC-grade methanol were purchased from Alfa Aesar (Ward Hill, MA, USA). Methylene chloride was obtained from Acros Organics (Fair Lawn, NJ, USA). Top Job bleaching solution was obtained from the local store. Deionized water (18 MΩ-cm) was prepared in our laboratory using a Purelab Ultra system from ELGA (Lowell, MA, USA). GD/X PVDF membrane filters (0.45 μm) were acquired from Whatman (Florham Park, NJ, USA). Strata SPE silica-2 sample (3 mL) tubes were received from Phenomenex (Torrance, CA, USA). Petri dishes (6 cm) were obtained from BD Falcon (Franklin Lakes, NJ, USA). Alltech Adsorbosil C18 column (4.6 × 150 mm, 5 µm) was purchased from Alltech Associates, Inc. (Deerfield, IL, USA). An Empty stainless steel tube (5 × 1.00 cm I.D. with 1.27 cm O.D.) and end fittings were received from Chrom Tech, Inc. (Apple Valley, MN, USA). OP50 and *Caenorhabditis elegans* N2 Bristol wild-type worm were obtained from *Caenorhabditis* Genetics Center (University of Minnesota, Minneapolis, MN, USA).

### 3.2. Preparation of Solutions

Propyl paraben was used as an internal standard. This solution was prepared by adding 0.0500 g of propyl paraben to a 50 mL volumetric flask and diluted to the mark with methanol. A stock solution was prepared by adding 0.00020 g each of tanshinone I and tanshinone IIA to a 10 mL volumetric flask. Then, 4.00 mL of dichloromethane was added into the volumetric flask. The volumetric flask was vortexed to obtain a homogeneous mixture. Then, 0.0100 g each of protocatechualdehyde, caffeic acid, and ferulic acid were added to the same volumetric flask and diluted to the mark with methanol. Calibrated standard solutions were prepared using both stock and internal standard solutions. The concentrations of the calibration solutions ranged from 0.002 to 1.00 mg/mL.

### 3.3. Subcritical Water Extraction of Salvia Miltiorrhiza

The extraction of *Salvia miltiorrhiza* was carried out using a home-made subcritical water extraction system, as shown in Figure 3. Both end fittings of a stainless steel extraction vessel were wrapped with Teflon tape for proper sealing. One end of the vessel was sealed with an end fitting first. Approximately 2 g (the actual weight was recorded to four decimal place) of *Salvia miltiorrhiza* (finely cut to small pieces, a few millimeters in length) was added to the stainless steel vessel. The void space of extraction vessel was filled with precleaned sand. The other end of the stainless steel vessel was then sealed with another end fitting. The loaded vessel was placed in an oven (HP gas chromatograph 5890 Series 2, Hewlett Packard, Avondale, PA, USA), as shown in Figure 3.

An ISCO model 260 D syringe pump (Lincoln, NE, USA) was used to supply 18 MΩ-cm water by opening V_1_ and closing V_2_ to fill the loaded vessel. Leak check of the extraction vessel was performed in the constant-pressure mode. It should be pointed out that a delay between the actual temperature of the extraction vessel and oven temperature was determined. The delay was 10 min for 75 °C, 12 min for 100 °C, 14 min for 125 °C, and 16 min for 150 °C. Static extraction was performed for 30 min after the delay time was compensated. A pressure of 15 to 25 atm was applied to keep hot water in the liquid state for all experiments. After 30 min of heating, approximately 10 mL of herbal extract was collected at 1 mL/min into a 25 mL glass vial by opening V_2_. Triplicate SBWE experiments were conducted at all temperatures.

### 3.4. Traditional Herbal Decoction of Salvia Miltiorrhiza

Approximately 2 g (the actual weight was recorded to four decimal place) of *Salvia miltiorrhiza* (finely cut to small pieces, a few millimeters in length) was added to a 50 mL glass beaker. Then, 10.00 mL of deionized water was added to it. The beaker was covered with a watch glass and heated up to boiling on a hot plate. Then, the temperature was adjusted to ensure the water kept boiling for 30 min. Triplicate THD experiments were conducted.

### 3.5. Sample Treatment

For characterization of SBWE water–herbal extracts on GC/MS, solid-phase extraction (SPE) was carried out using a silica phase cartridge and methanol as the elution solvent. At first, the silica cartridge was cleaned with approximately 5 mL of methanol followed by 10 mL of water. Then, the herbal extract was run through the silica cartridge and eluted using 1.00 mL of methanol into a 2 mL glass vial. Then, 30 µL of propyl paraben internal standard solution was added.

For HPLC analysis of tanshinone I and tanshinone IIA, liquid–liquid extraction was conducted. First, 1.00 mL of methylene chloride was added to each glass vial containing SBWE water–herbal extract. These vials were then sealed with aluminum-lined caps. These vials were vortexed to effectively mix the two phases. After separation of the two phases, the methylene chloride phase was transferred into an empty 5 mL glass vial. The same liquid–liquid extraction procedure for the SBWE water–herbal extract was repeated with another 1.00 mL of fresh methylene chloride. Again, the methylene chloride layer was removed and combined with the first fraction of the methylene chloride extract in the 5 mL vial. Then, 30.00 µL of propyl paraben internal standard was added to the methylene chloride phase.

To the aqueous phase of the herbal extract sample, 1.00 mL of methanol was added. Then, 300 µL of propyl paraben internal standard was added and mixed well. This sample was then filtered through a Whatman GDX filter into a glass vial for chromatographic analysis of protocatechualdehyde, caffeic acid, and ferulic acid.

### 3.6. HPLC Analysis

Please note that *Salvia miltiorrhiza* contains tens if not hundreds of compounds, and the five APIs investigated in this work are mixed with all other compounds in the extract. Therefore, we needed a standard method such as HPLC to achieve separation and quantification of our analytes to ensure the quality of this research. Thus, Shimadzu Nexera UFLC was employed for separation and quantification of *Salvia miltiorrhiza* extracts on an Alltech Adsorbosil C18 column using a methanol–water mixture as the mobile phase with 1.0 mL/min at ambient temperature. The eluents were detected at 254 nm.

### 3.7. GC/MS Analysis

In order to separate and identify the five APIs studied in this work, Agilent Technologies 6890N Network GC System (Santa Clara, CA, USA) coupled with a JEOL Ltd. JMS-GCmate II MS System (Tokyo, Japan) was employed for the characterization of SBWE extracts of *Salvia miltiorrhiza*. The GC separations were carried out on an Agilent HP-5MS (5% phenyl)-methylpolysiloxane (30 m × 0.250 mm, 0.25 μm film thickness) capillary column with 1.0 mL/min flow of a helium carrier gas. The sample volume was 1 µL, and it was injected using split mode by keeping the injector temperature at 250 °C. The GC/MS interface and the MSD ion chamber were set at 250 °C. The MS solvent delay time was 3 min. The GC oven temperature programming was as follows. The initial temperature was held at 30 °C for 3 min. Then, it was increased at 7.4 °C/min to 250 °C and maintained at 250 °C for 16 min. TSSPro Version 3.0 (Shrader Analytical and Consulting Laboratories, Inc., Detroit, Michigan, USA) was used for data acquisition and analysis.

### 3.8. Reproduction Studies on Caenorhabditis Elegans

A hermaphrodite, *Caenorhabditis elegans* N2 Bristol wild-type worm, was used for the reproduction assay to determine the reproduction rate of the herbal extracts. Synchronized L1 worms were cultured on NGM with *Escherichia coli* bacteria (OP50) as food. The NGM was supplemented with SBWE herbal extracts with a certain fold of dilution, which were obtained at extraction temperatures of 75, 100, 125, and 150 °C. Each treatment contained 15 worms with five biological replicates. These plates were incubated at 20 °C. After 30 h, these worms were washed off from each plate using M9 buffer into an Eppendorf tube. The tubes were then centrifuged twice with M9 buffer to wash worms of the herbal extract. Then, from each tube, about four worms were transferred to each plate already seeded with OP50 food. These plates were continuously monitored for egg laying. When worms started laying eggs, time was noted, and the plates were labeled as day 1 plates. These plates were incubated for another 24 h. The number of laid eggs was recorded for each day for three continuous days. The following equations were used to calculate reproduction inhibition and mortality rate.
(1)$$\mathrm{Reproduction}\;\mathrm{inhibition}=\frac{\mathrm{Control}\;\mathrm{average}\;\mathrm{reproduction}\;-\;\mathrm{Herbal}\;\mathrm{extract}\;\mathrm{average}\;\mathrm{reproduction}}{\mathrm{Control}\;\mathrm{average}\;\mathrm{reproduction}}\times 100$$
(2)$$\%\mathrm{Mortality}=\frac{\mathrm{Number}\;\mathrm{of}\;\mathrm{worms}\;\mathrm{died}}{\mathrm{Total}\;\mathrm{number}\;\mathrm{of}\;\mathrm{worms}}\times 100^{\;}$$

## 4. Conclusions

The research described in this work is different from any other existing herbal medicine preparation techniques. Unlike traditional herbal decoction, subcritical water extraction is conducted in a closed system where no loss of volatile active pharmaceutical ingredients to open environment occurs except analyte degradation at higher temperatures. Because temperatures other than 100 °C (the condition used in traditional herbal decoction) can also be employed in SBWE, subcritical water extraction at the optimized temperature should be more efficient in extracting active pharmaceutical ingredients from medicinal herbs than traditional herbal decoction. The higher extraction efficiency of SBWE should allow the subcritical water extraction time to be shortened, thus reducing the chance for degradation of the active pharmaceutical ingredients. These three factors should assure that optimized SBWE conditions would produce herbal medicine containing higher API concentrations than traditional herbal decoction. Compared with other industrial processes of making herbal medicine, subcritical water extraction eliminates toxic organic solvents. Therefore, it is not only environment friendly but also produces safer herbal medicine for patients.

Our results showed that the API quantity obtained by subcritical water extraction of *Salvia miltiorrhiza* increased by up to 4-fold by increasing the extraction temperature from 75 to 100 °C. They were then further enhanced by up to 26-fold with the increase of temperature from 100 to 125 °C, except for caffeic acid. When the extraction temperature was raised from 125 to 150 °C, API concentrations in SBWE extracts were further increased up to 4-fold, except for caffeic acid and protocatechualdehyde. Both caffeic acid and protocatechualdehyde might be degraded at 125 °C or higher. When comparing the tanshinone concentrations achieved by SBWE of *Salvia miltiorrhiza* with that obtained by THD, the SBWE extracts contained much higher tanshinone concentrations than the THD extracts.

The extraction temperature also plays an important role in the reproduction inhibition rate of the SBWE herbal extracts collected at four different temperatures. Reproduction inhibition evaluation of *Caenorhabditis elegans* revealed that the three-day average reproduction of worms decreased with increasing extraction temperature, while the reproduction inhibition rate increased from 6 to 46% when the SBWE temperature was raised from 75 to 150 °C. Please note that the reproduction inhibition of SBWE herbal extracts obtained at all temperatures from 75 to 150 °C was higher than that of traditional herbal decoction extracts.

In closing, besides subcritical water extraction being a more efficient technique than traditional herbal decoction in extracting anticancer agents from *Salvia miltiorrhiza,* SBWE herbal extracts also have higher reproduction inhibition rate than THD extracts according to our reproduction inhibition study. These findings demonstrate the potential of employing subcritical water extraction technique to develop high API-containing herbal medicine from *Salvia miltiorrhiza*.

## Figures and Tables

**Figure 1 molecules-26-01634-f001:**
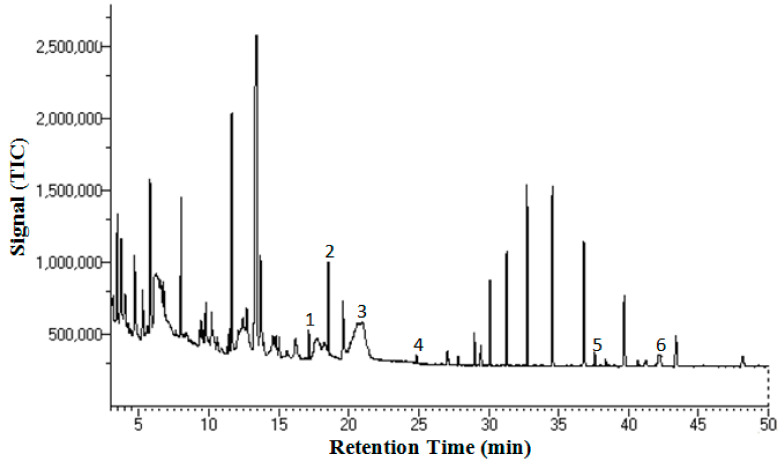
Total ion GC/MS chromatogram of a *Salvia miltiorrhiza* herbal extract obtained by subcritical water extraction (SBWE) at 150 °C for 30 min. Peak identification: 1, protocatechualdehyde; 2, propyl paraben (internal standard); 3, caffeic acid; 4, ferulic acid; 5, tanshinone IIA; and 6, tanshinone I.

**Figure 2 molecules-26-01634-f002:**
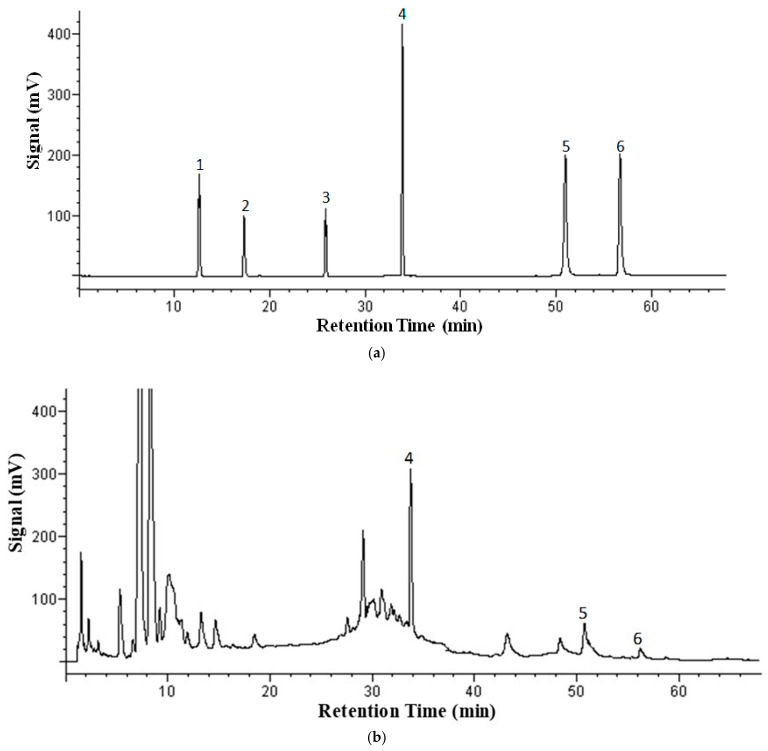
HPLC chromatograms of *Salvia miltiorrhiza* herbal extract obtained with 125 °C extraction temperature and evaluated in the Alltech Adsorbosil C18 column at ambient temperature. (**a**) Analyte standard solution; (**b**) methylene chloride phase; (**c**) water phase. Flow rate: 1.0 mL/min. UV detection: 254 nm. Mobile phase: A, 100 mM phosphoric acid in water; B, 100% methanol. Gradient: 0–4 min, 2% methanol; 4–8 min, 2–10% methanol; 8–23 min, 10–30% methanol; 23–32 min, 30–60% methanol; 32–43 min, 60% methanol; 43–49 min, 60–70% methanol; 49–61 min, 70–80% methanol; and 61–68 min, 80–2% methanol. Peak identification: 1, protocatechualdehyde; 2, caffeic acid; 3, ferulic acid; 4, propyl paraben; 5, tanshinone I; and 6, tanshinone IIA.

**Figure 3 molecules-26-01634-f003:**
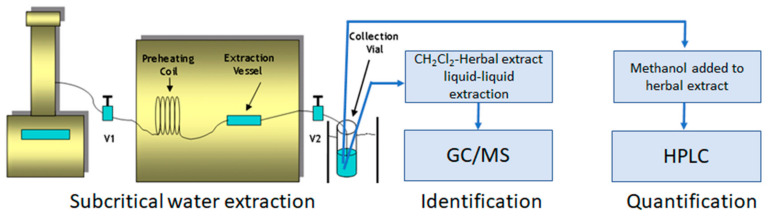
Block diagram of subcritical water extraction (V1 and V2 are needle valves) followed by GC/MS identification and HPLC quantification of APIs.

**Table 1 molecules-26-01634-t001:** Comparison of active pharmaceutical ingredient (API) concentrations found in *Salvia miltiorrhiza* obtained by traditional herbal decoction and subcritical water extraction.

Analyte	Concentration, μg/g (%RSD)^a^				
	Traditional Herbal Decoction, 100 °C	Subcritical Water Extraction			
		75 °C	100 °C	125 °C	150 °C
Protocatechualdehyde	19.6 (12.7)	11.4 (16.9)	29.1 (9.37)	701 (9.88)	1760 (5.26)
Caffeic Acid	51.6 (15.8)	47.3 (5.35)	57.3 (10.4)	48.2 (3.12)	16.1 (1)
Ferulic Acid	ND^b^	ND	ND	1.30 (20.3)	41.6 (13.9)
Tanshinone I	0.2 (10)	4.0 (2.4)	5.8 (7.4)	19.1 (13.4)	74.0 (3.43)
Tanshinone IIA	0.8 (20.1)	3.3 (1.3)	5.0 (13)	5.18 (16.1)	15.3 (12.7)

^a^ Triplicate measurements. ^b^ Not detected.

**Table 2 molecules-26-01634-t002:** Concentration of API in *Salvia miltiorrhiza* extracts used for reproduction study.

Analyte	Concentration, μg/mL	
	Traditional Herbal Decoction, 100 °C	Subcritical Water Extraction at 150 °C
Protocatechualdehyde	0.392	352
Caffeic Acid	1.03	3.22
Ferulic Acid	ND^a^	8.32
Tanshinone I	0.004	1.48
Tanshinone IIA	0.016	0.306

^a^ Not detected.

**Table 3 molecules-26-01634-t003:** Percentage reproduction inhibition and mortality of *Caenorhabditis elegans* after 30 h exposure to the 10 times diluted traditional herbal decoction and subcritical water extractions of *Salvia miltiorrhiza* at 75 to 150 °C.

Treatment NGM Plates	3-Day Average Production^a^ (%RSD)^b^	%Reproduction Inhibition (%RSD)^b^	%Mortality (%RSD)^b^
Control	111 (4.13)	0	0
THD	106 (5.35)	5 (20)	10 (20)
SBWE at 75 °C	104 (3.47)	6 (40)	17 (16)
SBWE at 100 °C	89 (12)	20 (20)	33 (15)
SBWE at 125 °C	67 (23)	40 (18)	42 (14)
SBWE at 150 °C	60 (34)	46 (12)	14 (21)

^a^ Total of eggs and larva average per worm over three days. ^b^ Five replicates.

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
