# Peer review of "Subcritical Water Extraction of Salvia miltiorrhiza"

_molecules, 2021, doi:10.3390/molecules26061634_

Round 1

Reviewer 1 Report

The manuscript addresses a relevant topic in the area of ​​natural products. However, the work has flaws that do not allow its publication in Molecules in the present form. The following are my notes.

1. The authors performed a reproduction assay against Caenorhabditis elegans, but sometimes they refer to it as a cytotoxic assay. No evaluation of the toxic action of the samples on the cells of the tested individuals was performed. Therefore, authors must explain why an assay made under these conditions can be called cytotoxic.

2. The authors present, as an advantage of the method employed, the fact that there is no loss of volatile vegetable constituents during subcritical extraction. However, they do not present any results that justify such a statement.

3. In Figure 3 only 4 peaks are marked, peaks 5 and 6 are missing.

4. Were the samples dried before biological testing? The authors must describe the concentration of the samples used in the biological assays (micrograms per milliliter, for example) in order for the results to be reproducible.

5. "The main reason for lower mortality of Caenorhabditis elegans with the cytotoxicity evaluation of herbal extract obtained at 150 ° C may be attributed to the less intake of high concentrated herbal extract through the skin of Caenorhabditis elegans."

From this sentence, I understand that the samples were not evaluated at the same concentration, as it is understood that the 150 ° C was more concentrated than the others. If this is true, the methodology used is not able to compare the "cytotoxic" activity of the different extractions, as they were not on equal terms.

6. "Another reason for the lower mortality of worms may be due to the degradation of compounds associated with the mortality of worms such as protocatechualdehyde at 150 ° C" (Lines 193-195).

In fact, it was not protocatechualdehyde that degraded at 150 ° C, but caffeic acid (Table 1).

7. Were the Reproduction Inhibition and Mortality assays performed in replicate? Why is there no standard deviation in the results shown in Table 2? If there were no replicates, the results of these assays cannot be considered.

8. Statistical analysis of the results of biological assays is necessary. For example: are there any differences between the Reproduction Inhibition values ​​of THD and SBWE-75 ° C?

9. Mathematical equations should be described in Materials and Methods, not in the legends of the tables.

Reviewer 2 Report

The manuscript submitted for review presents the comparison of the traditional herbal decoction with subcritical water extraction of Salvia Miltiorrhiza according to the extraction efficiency. Undoubtedly, the findings demonstrate the great potential of employing the subcritical water extraction technique. All advantages have been described in detail in the above paper. I appreciate their efforts and recognize them as the worth of the paper.
The authors applied HPLC and GC for identification and quantification of the chosen analytes. However, it is not clear for readers why they have applied these techniques, especially since both required different sample pre-treatment procedures. The authors need to explain their ideas in more detail.
Moreover-please-consider the following issues in improved submission:

-The introduction part should be shortened. Please focus only on data and do not explain obvious issues especially concerning the advantages of green chemistry.
-Please explain why two methods of identification have been applied HPLC and GC? Looking at chromatograms identification of all chosen extract components was possible only by GC/MS. 
-I do not see Fig.3a and 3b and 3c? I can see only Fig 3.
-Table 1 shows the comparison of Concentrations of analytes found in Salvia miltiorrhiza . There is no information about the method used for this quantification.
-Table 1- The authors wrote the following: Traditional Herbal
Decoction, 25 °C. What does the temperature of 25 °C mean? The decoction is conducted under 100 °C.
-Table 1 the %RSD values should possess the same number of significant figures as the measured parameters.
-Where is Fig.1?
-The paper concerns the quantitative analysis, so obligatory should present validation parameters. There is no information about the range of linearity, limits of detection and quantification, concentration of calibrated solutions. 
-Fig.4 should be deleted because illustrates the same data as in Table 1.
-The authors applied a handmade extraction system. Do they patent it? In my opinion, it will be difficult to repeat their experiments by others as well as prepare extracts by this methodology by consumers. Is it really necessary to produce your own handmade system? Why it is better than already existing ones? Explain, please.

Reviewer 3 Report

  1. Elements of scientific novelty should be presented in a detailed and convincing manner (in the last paragraph of the Introduction).
  2. I suggest that a diagram (scheme) presenting the used analytical procedures used in the study should be added to Methods section. 
  3. Because Authors declare that the developed procedure is green taking into account the principles of Green Analytical Chemistry, it will be perfect to present also this kind of results, and maybe green aspects of different approaches known from the literature should be also shortly discussed. Please consider at least on of the method: the Analytical Eco-Scale [1], Green Analytical Procedure Index [2].

    [1] A. GaÅ‚uszka, Z.M. Migaszewski, P. Konieczka, J. NamieÅ›nik, Analytical Eco-Scale for assessing the greenness of analytical procedures. Trends Anal. Chem., 2012, 37, 61–72.

    [2] J. PÅ‚otka-Wasylka, A new tool for the evaluation of the analytical procedure: Green Analytical Procedure Index, Talanta, 2018 In press DOI: https://doi.org/10.1016/j.talanta.2018.01.013

Round 2

Reviewer 1 Report

The modifications made by the authors improved the manuscript. However, there are issues that must be corrected before the manuscript can be considered for publication. In summary, there is a need for statistical analysis of the results and also to avoid statements that are not supported by the results.

Lines 25 and 128-129: Even in a closed system there may be a loss of the constituents of a plant due to degradation.

Lines 167-200: Comparisons between the concentrations obtained with the different extraction methods can only be performed through statistical analysis (for example, t-test). The same problem occurs with the discussion of the results of the reproduction assay (line 256).

About reproduction assay

Lines 417-418: As the biological assay was carried out with different concentrations of the extracts, it is not possible to compare the potency of the samples. Potency is defined as the amount of drug that is needed to produce a defined effect, commonly by estimating EC50 or IC50.

Reviewer 2 Report

The manuscript has been corrected and some explanations have been given, however, there are still some inconsistencies:

-Table 2- there is still 25 oC for decoction.

-Quantification is done by HPLC for all investigated compounds. Please explain why GC is necessary.

-Comparison with other extraction methods is necessary and required by most readers.

-Statistics for calibration curve should be given obligatory.

-At least partial validation is needed for technique utilizing handmade apparatus.

Reviewer 3 Report

I accept the current version of the submission

Author Response

This reviewer has already accepted our ms as it is.